# The influence of considering individual resistance training variables as a whole on muscle strength: A systematic review and meta-analysis protocol

**Philip M. Lyristakis**[1]*, **Daniel W. T. Wundersitz**[1], **Emma K. Zadow**[1], **George Mnatzaganian**[2,3], **Brett A. Gordon**[1]

**1** Holsworth Research Initiative, La Trobe Rural Health School, La Trobe University, Victoria, Australia, **2** Rural Department of Community Health, La Trobe Rural Health School, La Trobe University, Victoria, Australia, **3** The Peter Doherty Institute for Infection and Immunity, Melbourne, Victoria, Australia

* P.Lyristakis@latrobe.edu.au

**Funding:** The authors received no specific funding for this work.

## Abstract

Examinations of the effect of resistance training (RT) on muscle strength have attempted to determine differences between prescriptions, mostly examining individual training variables. The broad interaction of variables does not appear to be completely considered, nor has a dose-response function been determined. This registered (doi.org/10.17605/OSF.IO/EH94V) systematic review with meta-analysis aims to determine if the interaction of individual training variables to derive RT dose, dosing, and dosage can influence muscle strength and determine if an optimal prescription range exists for developing muscle strength. To derive RT dose, the following calculation will be implemented: number of sets × number of repetitions × number of exercises × exercise intensity, while RT dosing factors in frequency and RT dosage considers program duration. A keyword search strategy utilising interchangeable terms for population (adult), intervention (resistance training), and outcomes (strength) will be conducted across three databases (CINAHL, MEDLINE, and SPORTDiscus). Novel to the field of exercise prescription, an analytical approach to determine the dose-response function for continuous outcomes will be used. The pooled standardised mean differences for muscle strength will be estimated using DerSimonian and Laird random effects method. Linear and non-linear dose-response relationships will be estimated by fitting fixed effects and random effects models using the one-stage approach to evaluate if there is a relationship between exercise dose, dosing and dosage and the effect on muscle strength. Maximised log-likelihood and the Akaike Information Criteria will be used to compare alternative best fitting models. Meta regressions will investigate between-study variances and a funnel plot and Egger's test will assess publication bias. The results from this study will identify if an optimal prescription range for dose, dosing and dosage exists to develop muscle strength.

**Competing interests:** The authors have declared that no competing interests exist.

## Introduction

Resistance training (RT), any external resistance applied while exercising that can be quantified and modified to provide progression as a participant becomes stronger (adapted from Martyn-St James et al. 2006) [1], is prescribed to aid in rehabilitation, and improve quality of life and physical performance [2]. Resistance training provides a potent anabolic stimulus, which can result in increased muscle strength [2–5], muscle hypertrophy [2–4], and muscle power [5], along with reduced fat-mass [2, 4], elevated metabolic rate [2, 4], and improved glycaemic control [2, 4]. Resistance training guidelines have been developed to direct the attainment of muscle strength, hypertrophy, and power [6, 7]. These guidelines have been adapted for individuals across the lifespan, from children to older adults [8–11], and encompass the manipulation of training variables; exercise selection and order, repetition cadence, range of motion, rest intervals, weight load, frequency, sets and repetitions, to specifically stimulate mechanical and metabolic stress [12–18].

Exercise programming not absolutely conforming to guidelines has demonstrated increases in muscle strength [19–21]. Therefore, a consensus on how to best integrate specific and quantifiable training variables (sets, repetitions, intensity, frequency, and duration of the program) to develop muscle strength has not yet been formed. It is not yet known how these specific variables (sets, repetitions, intensity, frequency, and duration of a program) contribute to the total dose of RT as their interaction has not yet been determined. Moreover, there are no clear guidelines for how repetition cadence, program duration, rest intervals, or range of motion should be prescribed to elicit muscle strength, as research is yet to fully examine the interaction of these variables with others (i.e. sets, repetitions, intensity, and frequency).

Original research [12, 22, 23], and systematic reviews and meta-analyses [24, 25] have investigated how individual RT program variables (i.e. volume [sets and repetitions], intensity, and frequency) influence muscle strength. Most studies however, have only considered each training variable in isolation, making it difficult to determine if multiple RT variables interact to influence muscle strength or identify a variable that most contributes to the development of strength. Numerous original investigations [12, 22, 23, 26, 27] have assessed an individual training variable whilst manipulating other variables (e.g. sets, repetitions, or intensity) but these studies fail to consider the potential interaction of all variables. Therefore, it remains unclear whether a specific training variable (e.g. the number of sets, the number of repetitions, intensity, frequency, or duration) is more important than another, or if the interaction of variables results in larger muscle strength adaptations.

There are three potential ways in which the variables are likely to interact with one another; 1) RT dose considers the interaction of RT volume (number of sets x number of repetitions x number of exercises) and intensity, allowing an individual RT session to be considered; 2) RT dosing considers the interaction between RT volume, intensity, and frequency, allowing a week of RT sessions to be considered; and 3) RT dosage considers the interactions between RT volume, intensity, frequency, and the duration of the program intervention, allowing all RT sessions throughout the intervention period to be considered [28, 29]. Although Price et al. [29] considered the number of sets, repetitions, load, and frequency in their evaluation of RT dosage, this fails to consider the duration of a program which would incorporate all training sessions from the beginning until the end of an intervention. Another element that hinders the investigation of individual variables interacting is a lack of agreement in the definition of terms around RT dose. For example, RT volume has been calculated using different methods [26, 30, 31], and often these same calculations are labelled differently (i.e. training load, mechanical work, volume of training) [22, 32, 33]. Therefore, it is important to set out precise definitions.

## Resistance training volume

Resistance training volume has been defined as; the number of sets × repetitions [31], sets × repetitions × load$_{(\%1RM)}$ [33], sets × repetitions × load$_{(kg)}$ [22], weight × sets × repetitions for each exercise [34], or number of sessions × repetitions × sets [30]. The inclusion of 'number of sessions' by Pina et al. [30] indicates the total RT work over a program's duration, whereas including load describes the total work of a training session [35]. The method of calculating RT volume should encompass intra-session variables relating to the summation of performing repetitions (number of sets × number of repetitions × number of exercises). While it is possible that previous research has included the number of exercises completed as a consequence of the total number of sets, it is not clearly reported and therefore needs to be considered. For the purpose of this research, RT volume will be considered as the number of sets × number of repetitions × number of exercises. Considering this as a total number of repetitions (albeit throughout a week) RT volume greater than 250 repetitions per week has a significant effect on lower body strength regardless of absolute intensity [33, 36]. While moderate (5–9 sets per week) and high (>10 sets per week) volume RT may lead to greater strength gains regardless of the training status of participants [24]. However, the number of exercises was not considered by Ralston et al. [24] potentially underestimating the RT volume in their meta-analysis. Although the specific mechanisms responsible for an increase in muscle strength continue to be debated, it can be hypothesised that higher volume of training, which typically occurs at lower intensity, would develop strength primarily through mechanical tension, muscle damage, and metabolic stress [37, 38].

## Resistance training intensity

Resistance training intensity is typically defined as a percentage of maximal strength (e.g. % 1 repetition-maximum [RM]) [39], although a subjective scale of perceived effort referred to as rate of perceived exertion (RPE) is also used in practice [40]. Greater muscle strength gains have occurred with high-intensity ($\geq$ 80% 1RM) RT programs, while significant muscle strength can be achieved with low-moderate intensity ($\leq$ 60% 1RM) RT programs performed to muscle fatigue [25, 32, 33, 41]. It is possible that higher intensity RT might trigger increased motor unit recruitment and firing, and alter agonist-antagonist ratios to develop muscle strength [22]. When training protocols were matched for total work (sets × repetitions × load) between low-moderate intensities ($\leq$ 60% 1RM) and high intensity ($\geq$ 80% 1RM) programs, both are effective at increasing strength with a slightly greater effect observed with high intensity [32]. When intervention groups were work matched, a small 7–8% increase in strength favouring heavier loads was observed by both Csapo et al. [32] and Schoenfeld et al. [41] These results suggest that although significant muscle strength gains can be achieved with low-moderate intensities, higher intensities might be more important to stimulate maximal gains in muscle strength, particularly in trained individuals [41]. Higher intensities may produce greater muscle strength adaptations due to previously stated neural adaptations, while lower intensities might require additional repetitions (or volume) to supplement neural mechanisms with metabolic stress [42, 43]. It is also possible that lower intensity RT might require a greater contribution of slow-twitch type 1 muscle fibres and promote adaptations in the primary motor cortex, spinal cord or motor neuron [44, 45]. However, any summative value of different neurophysiological mechanisms for the development of muscle strength has not yet been investigated.

## Resistance training frequency

Resistance training frequency represents the number of RT sessions performed per week and per muscle group [46]. The frequency of RT sessions is an important component of resistance

exercise prescription, as the recovery time between sessions allows for muscle adaptation [7]. It is believed sufficient rest between RT sessions allows for the removal of metabolic by-products, replenishment of energy stores, and the initiation of tissue repair [47]. Fragala et al. [11] recommended a training frequency of two to three RT sessions per week, per muscle group, to provide an optimal stimulus to maximise increases in strength. This RT frequency has been reaffirmed by Feigenbaum et al. [7] who recommended a 48 hour rest between concurrent training sessions, which corresponds to three training sessions per week for individual muscle groups. It is entirely feasible though that training with lower intensities does not induce the same amount of muscle damage, and therefore requires less recovery time, but this has not yet been investigated. More frequent RT sessions could facilitate enhanced motor learning and as a result, increased muscle strength [48], independent of the proposed time for muscle adaptation.

While greater RT frequency is associated with larger muscle strength gain, this might be primarily influenced by RT volume, as Grgic et al. [17] observed no significant effect of RT frequency on muscle strength gains. Pina et al. [30] reported similar gains (19.5% versus 22.2%) in muscle strength in older women undertaking two and three RT sessions per week for 24-weeks, however, RT volume was not controlled between groups. As a result, it is not clear whether frequency or the RT volume accumulated with the additional training sessions contributed to the changes in muscle strength and the dosing of RT is worthy of additional investigation.

## Resistance training duration

Resistance training duration, the accumulation of weeks performing the RT program, has received minimal investigation, although re-testing of muscle strength at various points throughout an intervention has been reported. For instance, Pyka et al. [49] examined muscle strength changes in response to a 52-week RT intervention, where muscle strength was re-examined every 2-3-weeks. Findings from this study indicate that muscle strength rapidly increased in the first three months, then plateaued for the duration of the intervention [49]. The rapid increase in muscle strength in the first three months is likely due to neurophysiological mechanisms such as increased neural coordination and exposure to metabolic stress, while smaller long term changes could be associated with muscle hypertrophy [44]. The lack of good quality scientific investigation into RT program duration and the development of muscle strength means that it is currently not clear how long RT programs should be completed to optimise strength development through the potential interaction of different neurophysiological mechanisms.

## Resistance training dose

Because of the variation in definitions for RT variables, efforts to evaluate RT dose are limited. Dose-response relationships for individual RT variables (volume, intensity, frequency, and duration of training program) have been examined in older adults ($\geq$ 60 years) by Borde et al. [50] using standardised mean effects comparing one category to a control. Although this approach does not allow for an actual dose-response function, the duration of training, intensity, and total time under tension were identified as having a significant influence on muscle strength [50]. Borde et al. [50] suggested that time under tension initiated different metabolic changes as well as motor unit recruitment and firing, which could be important for developing strength. However, it is possible that these mechanisms are initiated through an increased volume with higher numbers of sets or repetitions of each exercise. A limitation that Borde et al. [50] identified was that the data available was insufficient to ascertain any potential

interactions between the individual variables. Systematic reviews and meta-analyses have yet to examine the influence of any interactions of RT variables on muscle strength by failing to calculate and compare dose, dosing, and dosage, due to differences in definitions along with failure to consider variables on a continuous scale.

### Does the interaction of specific resistance training variables to quantify the amount of training completed influence the dose-response function for muscle strength?

To date, it is not clear if there is a single most important RT variable, and if there is, which variable it is that most influences muscle strength. Further, it remains unclear if there is an optimum RT dose when the variables are considered as a whole. With the available research centred around individual RT variables, a holistic approach might be beneficial to investigate how the individual RT variables interact in the development of muscle strength, particularly given the apparent variations in neurophysiological mechanisms to derive increased strength. It is important to delineate how the dose (number of sets × number of repetitions × number of exercises × intensity), dosing (sets × repetitions × exercises × intensity × frequency) and dosage (sets × repetitions × number of exercises × intensity × frequency × duration) influences the development of muscle strength, and if a primary driver of this outcome can be identified. These combinations of variables have the potential to induce different neurophysiological mechanisms including, motor unit recruitment and firing, motor learning (specific fibre types), agonist-antagonist co-activation, metabolic stress, and increased cross-sectional area. However, it is unknown if these variables and potential mechanisms indeed interact or are summative to induce larger increases in muscle strength. Given that changes in training volume, training intensity and intervention duration have been identified as important factors by Borde et al. [50] and are likely to induce different mechanisms, it is hypothesised that there will be a dose-response effect that is magnified when the interaction of each training variable is considered. To investigate the stated hypothesis, methods have been developed to identify if a change in RT; 1) volume, 2) intensity, 3) frequency, 4) dose, 5) dosing, and 6) dosage is associated with a change in muscle strength in healthy adults.

### Methods

This systematic review protocol has been prospectively registered with Open Science Framework (OSF) [51], and is reported in accordance with the preferred reporting items for systematic reviews and meta-analyses protocol (PRISMA-P) guidelines (S1 File) [52]. A comprehensive keyword search strategy will be conducted across three databases; CINAHL, MEDLINE, and SPORTDiscus. Due to the interchangeable terms used to describe 'strength', 'muscle', 'weight training', and 'adults', a thorough search string using the population, intervention, comparison and outcomes (PICO) framework has been developed to capture as many relevant articles as possible (Table 1). Only data published in scientific journals after peer review will be included to minimise potential limitations and biases. The corresponding author of eligible studies will be contacted in the event insufficient information is available to calculate RT dose, dosing, or dosage. Databases will be searched from the earliest date possible until the 1st of December 2021.

Muscle strength data from eligible intervention groups will be compared with data from individuals receiving no exercise intervention or a placebo exercise intervention (i.e. stretching and/or mobility exercise) not designed to improve strength. Aerobic exercise only interventions have not been considered appropriate as a comparator to exclude any potential strength benefits from this mode of training, particularly in exercise-naïve populations. Concurrent

**Table 1. Search terms used to identify articles in CINAHL, MEDLINE, and SPORTDiscus.**

| Population | Intervention | Comparison | Outcomes |
|---|---|---|---|
| *Elderly* | *Resistance Training* | *Non-Exercise Control* | *Muscle Strength* |
| *Young Adult* | *Resistance Exercise* | *Placebo (stretching and mobility)* | *Muscular Strength* |
| *Older Adult* | *Weight Training* | *Non-Exercising Off-Season Athletes* | *Muscle Morphology* |
| *Middle-Age* | *Weight Lifting* | | *1 RM* |
| *Adult* | *Strength Training* | | *Repetition Max** |
| | *Free Weight* | | *Rep* Max** |
| | *High Intensity Training* | | *Physical Strength* |
| | *Progressive Resistance* | | *Force Development* |
| | *Physical Resistance* | | *Max* Strength* |
| | *Weighted Exercise* | | |
| | *Resistance Program* | | |
| | *Resistive Load* | | |
| | *Body Building* | | |
| | *Power Lifting* | | |
| | *Training Frequency* | | |
| | *Training Volume* | | |
| | *Training Load* | | |
| | *Training Intensity* | | |

Note–RM = repetition maximum

* = truncation.

exercise interventions (e.g. team sport and running) will be excluded as these interventions alongside RT may interfere with muscle strength adaptations [53]. Studies that include the prescription of abdominal exercises will be included, however, the actual abdominal exercises will be excluded from the calculation of dose, dosing, and dosage as the intensity of these exercises cannot be accurately quantified. These studies have been included as abdominal exercises are often prescribed, although not quantifiable, and it is not clear to what extent abdominal exercises may aid in maximal strength. Where an asterisk is used in the PICO search terms, this denotes a truncation; a technique that broadens a search to include various word endings and spellings. The search terms will be individually combined with the OR Boolean, before each construct will be combined using the AND Boolean.

## Study selection

This systematic review will include randomised controlled trials on populations aged 18 years and older, who are considered healthy and without a current diagnosed medical condition, unless they are deemed to meet the exclusion criteria (Table 2). Specifically, for exclusion criteria five, supplementary interventions will be considered as concurrent exercise or nutritional interventions in addition to the quantifiable RT program. From the eligible studies, information on individual training variables will be extracted and used to calculate the dose, dosing, and dosage of RT and the influence these variables have on muscle strength. The particular interest for this study is dynamic muscle strength, which must be measured using RM assessments or isokinetic dynamometer testing. Peak torque at isokinetic velocities between 60–120˚/sec are indicative of muscle strength, where 180˚/sec or higher is aligned with muscle power [54]. As a result, studies using isokinetic dynamometer testing above 120˚/sec will be excluded. The identified exclusion criteria will be applied to each identified article title and

**Table 2. The exclusion criteria to remove ineligible studies.**

| Number | Exclusion Criteria |
| --- | --- |
| 1 | Written in languages other than English. |
| 2 | Reviews, commentaries, conference abstracts, posters, non-peer reviewed articles, and studies other than randomised controlled trails or those that do not provide original data. |
| 3 | Non-human studies. |
| 4 | Participants under the age of 18, or with a current diagnosed medical condition. |
| 5 | Resistance training programs lasting less than 4-weeks, or those that do not have one intervention and a no-exercise control intervention not receiving any supplementary intervention. |
| 6 | Studies that do not provide a measure of dynamic muscular strength (i.e. 1–10 RM test, dynamometer) pre- and post-intervention. |
| 7 | No quantifiable measures of dynamic bilateral RT provided for at least one RT variable; intensity, frequency, and volume (sets, repetitions, and number of exercises). |

abstract by two independent investigators with a third investigator to provide a majority decision when conflicting views are reported. The same approach will be used for full-text screening.

## Assessment of methodological and outcome quality

The TESTEX Scale is a validated tool for the assessment of study quality and reporting in exercise training studies and will be used to asses for the risk of bias [55]. This tool encompasses 12 criterion assessments with a maximum score of 15 and takes into account eligibility criteria and concealment of allocation. Subsequent blinding of patients and therapists is nearly always unachievable in exercise training studies and does not reduce the assessment of quality [55]. For completeness, methodological quality of studies will also be evaluated using the Pedro checklist [56]. Quality of outcomes will be evaluated using the grading of recommendations, assessment, development, and evaluations (GRADE) approach [57]. Methodological quality and quality of outcomes will be evaluated independently by two independent investigators and any disagreements will be resolved through discussion and consensus with a third investigator.

## Data extraction

Participant demographic data (i.e. age, sex, and training status) and specific information relating to program interventions will be extracted from eligible studies into a Microsoft Excel spreadsheet. These data include; the number of sets, number of repetitions, number of exercises, and sessions per week in which the program intervention was conducted, the absolute and/or relative intensity will also be extracted from eligible articles. Where a range is presented or variables progress from one unit to another throughout the program, the median will be entered into the data extraction spreadsheet. The median was chosen as wide ranges in resistance training intensity (i.e. 20% 1RM to 100%1RM) can be applied, along with variation in how it is progressed throughout an intervention. If the mean was to be used instead of the median, it is possible that intensity could be under- or over-estimated. Utilising the median will provide the central tendency for skewed number distribution. Pre- and post-testing results of muscle strength assessments will be extracted from eligible studies. If mean and standard error or mean and 95% confidence interval are reported, they will be converted to mean and standard deviation [58]. Where a figure illustrates pre- and post-intervention muscle strength testing, Microsoft Visio (Version 16.0.13929.20206, 2016, Microsoft Corporation, Redmond, Washington, USA) will be used to determine the mean and standard deviation. Data will be extracted from eligible articles by an independent investigator. Accuracy will be assured by

randomly selecting 10% of eligible articles to have data extracted by a second independent investigator. If the results vary by more than 5% in more than 25% of the selected articles, data from all articles will be extracted by a second investigator. Any discrepancies in data will be resolved through consultation with a third investigator.

The extracted data will be published and freely accessible via FigShare after the data have been fully curated and published.

## Data treatment

Data pertaining to resistance training variables extracted as described, will be subjected to several treatment approaches to derive the proposed composite values of dose, dosing, and dosage. The specific exercise prescribed in each study will be identified and the relevant number of sets and repetitions will be recorded, along with the prescribed intensity (load). For each exercise, the number of sets, number of repetitions and prescribed intensity will be multiplied to derive the volume of work for each specific exercise. Following this, the volume of work for each specific exercise will be summed to derive the training dose. Once the training dose is derived, this will be multiplied by the weekly training frequency to derive dosing. The dosing will be multiplied by the training duration (weeks) to derive dosage. If sufficient data is not available for each data treatment stage, the article will be excluded from subsequent analyses.

## Data analysis

The main study outcome measure will be the standardised mean difference (SMD) in muscle strength estimated as the mean difference of the cases and controls divided by the standard deviation of the overall population in the study. The pooled standardised mean differences (SMD) for muscle strength data (continuous outcome measure) will be estimated using the DerSimonian and Laird random effects method [59]. Linear and non-linear dose-response relationships will be estimated by fitting fixed effects and random effects models using the one-stage approach [60] to compare the SMD for strength against the individual prescribed training variables (volume, intensity, frequency) and composite training prescriptions (i.e. dose, dosing, dosage). Maximised log-likelihood and the Akaike Information Criteria (AIC) will be used to compare alternative models to identify the best fitting model. The proposed analysis allows for curvilinear relationships by using transformations of the dose (e.g. splines and polynomials) and accounts for between study heterogeneity in the true dose-response relationships by adding random effects in the regression coefficients of the transformation of the dose (individual RT variable along with calculated dose, dosing, and dosage). Random effect models also assist in controlling for unobserved heterogeneity when the heterogeneity is contact over time and not correlated with independent variables [59]. Meta regressions will be constructed to investigate and quantify the proportion of between-study variance explained by known study variables including age, sex, training status, body musculature, muscle strength adaptations and risk of bias. Publication bias will be assessed using funnel plots and Egger's tests. Asymmetry will be considered if the $P < 0.10$ for Egger's test, indicating potential publication bias [61].

After the primary analyses have been completed, a series of sub-analyses will be conducted, if possible, to examine the effect of age (young vs. middle-aged vs. elderly), sex (male vs. female), training status (trained vs. untrained), and assessed body musculature (lower vs. upper-body) on muscle strength adaptation following RT. Sensitivity analyses will be conducted by risk of bias, in which the analyses will be re-run while excluding poor quality studies.

The analyses will be conducted using Stata/SE 16.1.

## Discussion

This systematic review will meticulously explore the available evidence on the influence of dose, dosing, and dosage of RT on muscle strength. Although previous attempts to evaluate the dose-response of individual RT variables using standardised mean effects have identified key variables of interest, there has not been a consistent approach to determining the overall dose, dosing and dosage obtained through the interaction of variables [50]. The approach proposed will utilise dose as a continuous variable instead of categorising variables/outcomes to identify if one category is different to another. By gathering, analysing, and synthesising the information about the number of sets, number of repetitions, number of exercises, and RT intensity, frequency, and duration of the RT program on a continuous scale and evaluating if the interaction of these variables influences muscle strength differently, this study might offer new directions for practice and future research. Providing an optimal range of RT dose, dosing, and dosage to develop muscle strength might be possible if a Poisson distribution is identified. A Poisson distribution would offer clarity on how volume, intensity, frequency, and duration interact in the development of muscle strength, which could be important for practitioners. The outcome of this systematic review may identify a key training variable or combination of variables that are superior for the development of muscle strength. The results from this systematic review are also likely to assist health and fitness professionals, strength and conditioning coaches, and rehabilitation staff to suggest appropriate exercise prescription for a range of populations, regardless of the setting and facilities available, with variables able to be manipulated to maximise outcomes. This may be particularly important in settings with low equipment availability (e.g. rural and regional locations) where it might be less feasible to complete high-intensity resistance training.

## Supporting information

**S1 File. PRISMA-P 2015 checklist.**
(DOCX)

## Acknowledgments

The authors acknowledge the Bendigo Tertiary Education Anniversary Foundation and La Trobe Holsworth Research Initiative's support of Dr Wundersitz's research.

## Author Contributions

**Conceptualization:** Philip M. Lyristakis, Daniel W. T. Wundersitz, Emma K. Zadow, George Mnatzaganian, Brett A. Gordon.

**Formal analysis:** George Mnatzaganian.

**Methodology:** Philip M. Lyristakis, Daniel W. T. Wundersitz, Emma K. Zadow, George Mnatzaganian, Brett A. Gordon.

**Supervision:** Philip M. Lyristakis, Daniel W. T. Wundersitz, Emma K. Zadow, Brett A. Gordon.

**Writing – original draft:** Philip M. Lyristakis.

**Writing – review & editing:** Daniel W. T. Wundersitz, Emma K. Zadow, George Mnatzaganian, Brett A. Gordon.

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
