## [Decision Letter · Decision Letter 0]

4 Oct 2021

PONE-D-21-23722Study Protocol for a Systematic Review with Meta-Analysis: The Influence of Resistance Training Dose, Dosage and Dosing on Muscle Strength.PLOS ONE

Dear Dr. Philip Lyristakis,

Thank you for submitting your manuscript to PLOS ONE. After careful consideration, we feel that it has merit but does not fully meet PLOS ONE’s publication criteria as it currently stands. Therefore, we invite you to submit a revised version of the manuscript that addresses the points raised during the review process.

Two experts in the field evaluated the study protocol. They have provided distinctive feedback (i.e., one focused on methods and the other on the study rationale) and evaluation (i.e., positive and negative recommendations), as noted by reading their reports. Although the study protocol has merit, the reviewers indicated several issues that should be addressed before the protocol is suitable for publication. Therefore, the reviewers asked for more clarification, particularly in the theoretical rationale supporting the systematic review/meta-analysis and the methods used.

We look forward to receiving your revised manuscript.

Kind regards,

Samuel Penna Wanner, Ph.D.

Academic Editor

PLOS ONE

Additional Editor Comments (if provided):

The second reviewer has evaluated the study protocol negatively. Despite this unfavorable evaluation, I understand that the authors still have time to improve this protocol as much as reviewers require (i.e., I understand that it is the idea of peer-reviewing a study protocol). Therefore, I am looking forward to receiving an improved version of this study protocol. Please consider carefully each comment made by reviewers and change the protocol accordingly.

Please also check the following minor points:

1- There is no need to place a comma between the authors’ name and the phrase “et al.”. For example, lines 73, 84, 95, 107, and 108.

2- Lines 178 – 181. These two sentences were written in the past tense. I have the impression that they will sound better if written in the future tense.

3- The date for database search should be updated to a date close to when the study protocol will be submitted.

Reviewers' comments:

Reviewer's Responses to Questions

**Comments to the Author**

1. Does the manuscript provide a valid rationale for the proposed study, with clearly identified and justified research questions?

Reviewer #1: Yes

Reviewer #2: Partly

2. Is the protocol technically sound and planned in a manner that will lead to a meaningful outcome and allow testing the stated hypotheses?

Reviewer #1: Yes

Reviewer #2: Partly

3. Is the methodology feasible and described in sufficient detail to allow the work to be replicable?

Reviewer #1: Yes

Reviewer #2: No

4. Have the authors described where all data underlying the findings will be made available when the study is complete?

Reviewer #1: No

Reviewer #2: No

5. Is the manuscript presented in an intelligible fashion and written in standard English?

Reviewer #1: Yes

Reviewer #2: Yes

6. Review Comments to the Author

You may also provide optional suggestions and comments to authors that they might find helpful in planning their study.

Reviewer #1: Overall, the manuscript is well written and designed. To determine if the interaction of individual training variables to derive RT dose, dosing, and dosage can influence muscle strength and determine if an optimal prescription range exists for developing muscle strength. But I have some questions that I think are important to be answered:

1- An interesting series of secondary analyzes were suggested, but did you also consider comparing the responses of men and women?

2- Among the outcomes described is Muscle Morphology. Do you intend to evaluate Muscle Morphology? Because the direction of the entire manuscript suggests that only Muscle Strength will be evaluated. If they intend to evaluate Muscle Morphology, the manuscript needs to make this clearer.

3- I suggest using the GRADE tool (Grading of Recommendations Assessment, Development and Evaluation) to evaluate levels of evidence for each of the analyzes you intend to do

4- The PROSPERO record number is in the abstract, please also add it in the text body.

5- Please add information suggested in the Plos One submission guideline:

• where and when the data will be made available. See our Data Availability policy for more.

• A completed PRISMA-P checklist must be provided as a supporting information (SI) file. See PRISMA-P Explanation and Elaboration for more information on completing your checklist.

Reviewer #2: Study Protocol for a Systematic Review with Meta-Analysis: The Influence of Resistance Training Dose, Dosage and Dosing on Muscle Strength.

Dear editor, the authors of the study in question propose to carry out a systematic review with meta-analysis to understand the effect of the interaction between the variables volume (series x repetitions x number of exercises) x intensity: dose; volume x intensity x frequency: dosing; volume x intensity x frequency x duration: dosage; in the maximum dynamics strength response after a resistance traning. A second objective would be to indicate an optimal prescription for increasing strength based on the results of these interaction.

There are already studies (i.e., systematic reviews) that individually analyzed intensity, volume, frequency and duration, and quite few studies that have analyzed the interaction of variables in the strength response. The reason to conduct a study should not be by the number of studies available in a determined theme, but by the reasoning, whether in biomechanics, physiology or any other field, which allows sustaining the study hyphothisis, and further the execution of a study.

The present study does not present any neurophysiological basis that justifies the combined analysis of the variables mentioned in order to increase maximum strength, and contribute to a more assertive prescription. In addition, the reason for performing three analyses: dose, dosage and dosage instead one, two or even more, was not discussed. It would be interesting to clarify the advantages of each analysis. Also, the Prospero ID (ID number: 1922649) was not found (https://www.crd.york.ac.uk/prospero/) as announced.

The basis for investigating either interaction is weak and requires additional information. To illustrate, the authors used more than one page in the introduction to conceptualize the variables. This approach is necessary, but brief, being important to substantiate/hypothesize why and how the interaction of these variables allows for a greater increase in the strength production after an intervention.

The authors proprose to review studies without target subject or body members (upper or lower), where the main inclusion criteria was the maximum dynamic strength been evaluated before and after an intervention from three databases (CINAHL, MEDLINE, and SPORTDiscus). This give me the impression of unfeasibility due to the extensive number of studies. I believe that without a direction (i.e upper limbs or any specific exercise or even population) this project is unlikely to be carried out.

Greater statistical detail is necessary, as it lacks information to explain how the data will be treated to answer the study questions, and achieving the objectives proposed.

In the discussion it is said that the results can be the key to a prescription of a defined variable or by combination. However, other important variables were left out, for example: range of motion and pause. I believe that the text should be a little more conservative and direct the discussion by the possible results found and not extrapolate in the sense of formulating an ideal prescription based only on the investigated variables.

For all the above aspects, I believe that the article is not fit for publication. Other information is provided in the attached document, addressed to the authors.

7. PLOS authors have the option to publish the peer review history of their article (what does this mean?). If published, this will include your full peer review and any attached files.

Reviewer #1: **Yes: **Lucas Rios Drummond

Reviewer #2: No

---

## [Author Response · Author response to Decision Letter 0]

19 Nov 2021

Editor Comments:

1. There is no need to place a comma between the authors’ name and the phrase “et al.”. For example, lines 73, 84, 95, 107, and 108.

Thank you for identifying this, we have amended these issues throughout the doucment.

2. Lines 178 – 181. These two sentences were written in the past tense. I have the impression that they will sound better if written in the future tense.

We have amended this section and now reads in future tense.

3. The date for database search should be updated to a date close to when the study protocol will be submitted.

We appreciate the desire to have an up to date search. As this is projected to be a large review (i.e. a large number of articles eligible for inclusion) we were proposing that no further updates of the search beyond the date that we originally indicated would occur. However, we can also see benefit of ensuring the search is as up to date as possible. Therefore, we will update the search on 1st of December, 2021 and not conduct any further search updates to ensure the search is reproducible. This has been added in the manuscript on Page 10, Lines 215-216 (of the unmarked manuscript).

 

Reviewer One:

Overall, the manuscript is well written and designed. To determine if the interaction of individual training variables to derive RT dose, dosing, and dosage can influence muscle strength and determine if an optimal prescription range exists for developing muscle strength. But I have some questions that I think are important to be answered.

Thank you for this positive outlook of our study design. We have responded to specific questions below. 

1. An interesting series of secondary analyzes were suggested, but did you also consider comparing the responses of men and women?

Thank you for this suggestion. We are anticipating many studies that will be eligible for inclusion in this systematic review and meta-analysis. As such we have included in the secondary analyses that dose-response effects between males and females will be assessed if sufficient data is available. 

Page 16, Lines 323-326: 

“After the primary analyses have been completed, a series of sub-analyses will be conducted, if possible, to examine the effect of age (young vs. middle-aged vs. elderly), sex (male vs. female), training status (trained vs. untrained), and assessed body musculature (lower vs. upper-body) on muscle strength adaptation following RT.”

2. Among the outcomes described is Muscle Morphology. Do you intend to evaluate Muscle Morphology? Because the direction of the entire manuscript suggests that only Muscle Strength will be evaluated. If they intend to evaluate Muscle Morphology, the manuscript needs to make this clearer.

We do not intend to evaluate muscle morphology, this term was used in the PICO (Table 1) “outcomes” which do not depict the outcomes proposed for this study. This outcome was included in the search strategy as studies examining muscle morphology may also evaluate muscle strength despite it not being a primary outcome. As such, this may lead to additional eligible studies that would be included in our systematic review and meta-analysis. As a result, no change has been made to the manuscript.

3. I suggest using the GRADE tool (Grading of Recommendations Assessment, Development and Evaluation) to evaluate levels of evidence for each of the analyzes you intend to do.

It may have been missed by the reviewer that we had stated in the manuscript under the subheading “Assessment of Methodological and Outcome Quality (Risk of Bias)” that quality of outcomes will be evaluated using the GRADE tool. The quality of individual studies included in the review and to assist with formulating GRADE outcomes will be determined by Pedro and Testex. To improve the clarity of this section and not to inadvertantly reduce the emphasis of the quality of outcomes, we have removed “(Risk of Bias)” from the sub-heading. This subheading now reads as: 

“ Assessment of Methodological and Outcome Quality” and can be found on Page 13, Line 255.

4. The PROSPERO record number is in the abstract, please also add it in the text body.

When we originally submitted the protocol manuscript, we had submitted the protocol for registration with PROSPERO. Since then, PROSPERO identified that we were not investigating a clinical population and therefore decided to cancel the registration. To clarify, the methods presented in this manuscript were not identified as an issue, only that we were not investigating a clinical population. Our protocol has since been registered with The Open Science Framework (OSF). The information has been revised in the abstract and included in the methods. 

Page 2, Line 18-19: 

“(doi.org/10.17605/OSF.IO/EH94V) systematic review with meta-analysis...” 

Page 10, Line 205-206: 

“This systematic review protocol has been prospectively registered with Open Science Framework (OSF),[51]...”

Please add information suggested in the Plos One submission guideline:

5. Where and when the data will be made available. See our Data Availability policy for more.

There is no underlying data available to share for this protocol paper. Therefore, at this time we can not share any data. Any data generated as a result of this review is intended to be published via FigShare. The location of this data will be made available when they have been fully curated and the findings are published. 

Page 14, Lines 288-289:

“The extracted data will be published and freely accessible via FigShare after the data have been fully curated and published.”

6. A completed PRISMA-P checklist must be provided as a supporting information (SI) file. See PRISMA-P Explanation and Elaboration for more information on completing your checklist.

We believe in being open and transparent in science. In our initial submission we included a PRISMA-P checklist but we have referred the reader to this now in the manuscript.

Page 10, Lines 205-208:

“This systematic review protocol has been prospectively registered with Open Science Framework (OSF),[51] and is reported in accordance with the preferred reporting items for systematic reviews and meta-analyses protocol (PRISMA-P) guidelines (supplementary file 1).[52]”

 

Reviewer Two:

Dear editor, the authors of the study in question propose to carry out a systematic review with meta-analysis to understand the effect of the interaction between the variables volume (series x repetitions x number of exercises) x intensity: dose; volume x intensity x frequency: dosing; volume x intensity x frequency x duration: dosage; in the maximum dynamics strength response after a resistance traning. A second objective would be to indicate an optimal prescription for increasing strength based on the results of these interaction.

1. There are already studies (i.e., systematic reviews) that individually analyzed intensity, volume, frequency and duration, and quite few studies that have analyzed the interaction of variables in the strength response. The reason to conduct a study should not be by the number of studies available in a determined theme, but by the reasoning, whether in biomechanics, physiology or any other field, which allows sustaining the study hyphothisis, and further the execution of a study.

We agree with the reviewer in that there have been systematic reviews that have analysed individual variables of resistance exercise prescription and some that have tried to consider their interaction. However, in all of the reviews that we are aware of, they have categorised the variables and compared one category to another. In the systematic review by Borde et al. the authors noted that it was not possible to consider how the variables interacted. In this review, we are proposing the most comprehensive approach to integrating variables that we can identify. We are also proposing not to simply compare groups or categories of exercise prescription but to conduct a dose-response analysis using two continuous variables using an approach that has recently become available. Most dose-resonse analyses use a categorical and continuous variable. Although this does not specifically provide biomechanical or physiological reason, we have added some text throughout the introduction to provide an overview of the possible neurophysiological responses that might contribute to increased strength, noting that the mechanisms of increasing strength are still debated.

2. The present study does not present any neurophysiological basis that justifies the combined analysis of the variables mentioned in order to increase maximum strength, and contribute to a more assertive prescription. In addition, the reason for performing three analyses: dose, dosage and dosage instead one, two or even more, was not discussed. It would be interesting to clarify the advantages of each analysis. Also, the Prospero ID (ID number: 1922649) was not found (https://www.crd.york.ac.uk/prospero/) as announced.

We have added some text throughout the introduction to provide some possible neurophysiological mechanism to contribute to increased muscle strength. Although this on its own probably is not sufficient to satisfy this reviewer, we have also identified in the introduction that the neurophysiological mechanisms for increasing muscle strength are still being debated. It is important to note though, that this review nor other reviews, is designed to identify potential mechanisms of increased strength, but to see if the strength response is magnified when the resistance training prescription variables are considered as a whole. While it is important to consider and hypthosise about potential mechanisms and reasons for why there might be an interaction of variables (which we have now included), we think it is just as important to determine if the variables interact. If the finding indicates that interaction is occuring, further research on mechanisms can then be undertaken. 

The reason for conducting the series of meta-analytical dose-response functions is to systematically add individual variables to the interaction and observe if the dose-response function changes. This will assist in determining if there is any one or a combination of variables that contribute most to the increased muscle strength that is a known outcome associated with resistance training. Although we are not aware of a statistical approach that will allow for the dose-response functions to be statistically compared, it will be possible to plot the curves from each meta-analysis on the same axes to demonstrate if any magnification in response is occuring.

The Prospero ID was registered. However, since the original submission of this protocol manuscript Prospero re-examined the registration and identified that no clinical population was being examined and therefore it was not suitable for registration with Prospero. Importantly Prospero did not raise concerns about the method, simply stating that the population was not appropriate to register the work with Prospero. We have now registered the protocol with open science framework (OSF) and as noted in response to Reviewer one, included this in both the abstract and the methods.

Page 2, Lines 18-19: 

“(doi.org/10.17605/OSF.IO/EH94V) systematic review with meta-analysis...” 

Page 10, Lines 205-206: 

“This systematic review protocol has been prospectively registered with Open Science Framework (OSF),[51]...”

3. The basis for investigating either interaction is weak and requires additional information. To illustrate, the authors used more than one page in the introduction to conceptualize the variables. This approach is necessary, but brief, being important to substantiate/hypothesize why and how the interaction of these variables allows for a greater increase in the strength production after an intervention.

We have added some text to the introduction to strengthen the rationale, specifically trying to address potential neurophysiological mechanisms. However, the evidence for mechanisms to increase muscle strength as a result of resistance training are equivocal. We have tried to clarify other elements relating to the points above from this reviewer and believe that there is now a strong rationale depicted for taking this approach.

4. The authors proprose to review studies without target subject or body members (upper or lower), where the main inclusion criteria was the maximum dynamic strength been evaluated before and after an intervention from three databases (CINAHL, MEDLINE, and SPORTDiscus). This give me the impression of unfeasibility due to the extensive number of studies. I believe that without a direction (i.e upper limbs or any specific exercise or even population) this project is unlikely to be carried out.

The reviewers concern is noted. However, I would encourage the reviewer not to judge feasibility without knowing the extent of the research team or any resources that might be at our disposal. We agree that not specifying a target population or specific body area will result in a very large search result. However, to be able to calculate an accurate dose-response, it is critical to have a large data set. We fully expect to have a much larger number of eligible articles than is typically identified from a more specified and refined search, but fully believe that we have capacity to complete this review. It is also partly the purpose for publishing the review protocol, to demonstrate our committment to completing this and acknowledging that the findings might take longer to arrive at and publish than a typical systematic review might.

5. Greater statistical detail is necessary, as it lacks information to explain how the data will be treated to answer the study questions, and achieving the objectives proposed.

As recommended, we have elaborated further on our statistical methods. The following has been added to the manuscript.

Page 2, Lines 33-35:

“Meta regressions will investigate between-study variances and a funnel plot and Egger’s test will assess publication bias”

Page 15, Lines 302-306:

“The main study outcome measure will be the standardised mean difference (SMD) in muscle strength estimated as the mean difference of the cases and controls divided by the standard deviation of the overall population in the study. The pooled standardised mean differences (SMD) of the extracted muscle strength data (continuous outcome measure) will be estimated using the DerSimonian and Laird random effects method [59].”

Page 16, Lines 316-322:

“Random effect models also assist in controlling for unobserved heterogeneity when the heterogeneity is contact over time and not correlated with independent variables.[59] Meta regressions will be constructed to investigate and quantify the proportion of between-study variance explained by known study variables including age, sex, training status, body musculature, muscle strength adaptations and risk of bias. Publication bias will be assessed using funnel plots and Egger’s tests. Asymmetry will be considered if the P < 0.10 for Egger’s test, indicating potential publication bias.[61]”

Page 16, Lines 326-327:

“Sensitivity analyses will be conducted by risk of bias, in which the analyses will be re-run while excluding poor quality studies.”

In addition, we have added a new subheading “Data Treatment” which can be found on Pages 14-15, Lines 290-300, and reads as:

“Data pertaining to resistance training variables extracted as described, will be subjected to several treatment approaches to derive the proposed composite values of dose, dosing, and dosage. The specific exercise prescribed in each study will be identified and the relevant number of sets and repetitions will be recorded, along with the prescribed intensity (load). For each exercise, the number of sets, number of repetitions and prescribed intensity will be multiplied to derive the volume of work for each specific exercise. Following this, the volume of work for each specific exercise will be summed to derive the training dose. Once the training dose is derived, this will be multiplied by the weekly training frequency to derive dosing. The dosing will be multiplied by the training duration (weeks) to derive dosage. If sufficient data is not available for each data treatment stage, the article will be excluded from subsequent analyses.”

6. In the discussion it is said that the results can be the key to a prescription of a defined variable or by combination. However, other important variables were left out, for example: range of motion and pause. I believe that the text should be a little more conservative and direct the discussion by the possible results found and not extrapolate in the sense of formulating an ideal prescription based only on the investigated variables.

It is indeed possible that other variables that we are not considering in this review could contribute to the development of strength. However, the evidence for the importance of these variables and how to use them in prescription is still being developed. For this reason we don’t beleive that these additional variables can be quantified as part of an interaction of variables to create a composite outcome. Despite not including these variables, we have changed the discussion to be more conservative about the potential outcomes of this review by specifying the training variables investigated.

7. For all the above aspects, I believe that the article is not fit for publication. Other information is provided in the attached document, addressed to the authors.

In responding to the above aspects and providing additional rationale we hope that you now believe the manuscript of our review protocol to be fit for publication. Additional evidence of this is provided by responding to the below comments and associated changes to the manuscript. We strongly believe that further consideration of how individual variables interact is important to see if it is possible to simply modify a single variable or if multiple variables should be manipulated in unison to encourage muscle strength development.

8. Dear authors, below are my comments. In opinion the work needs major improvement and should not continuing in the review. The work is well written, but the foundation for the execution/reproducibility of the project is little sustaintable. It would be interesting if the authors presented minimal mechanistic bases to justify the analysis of the interaction of variables. There are several studies that analyzed each of the variables in isolation, so why would it be necessary to carry out a combined analysis? What supports this analysis? And why should it be done from the perspective of dose, dosing, and dosage? What each of these analyzes contribute to the training prescription? To illustrate, the authors used more than one page in the introduction to conceptualize the variables (Ln 80-137). This approach is necessary, but brief, being important to substantiate/hypothesize why and how the interaction of these variables allows for a greater increase in strength production after an intervention. 

Thank you for the positive comment relating to the writing. We appreciate the reviewers concern around the mechanistic bases, and as previously outlined, have added elements of this throughout the introduction to identify that different variables could be inducing strength changes through various mechanism, and it is unclear if these are additive. Although we considered taking a briefer approach to the introduciton, we felt it important to clearly outline and define each of the variables of interest (while acknowledging the potential role of other variables) and describing the potential related mechanisms. Therefore the introduction has been expanded.

9. It would be interesting to clarify which are the dependent and independent variables of the study and how the statistical procedures will contribute to the establishment of dose, dosage, and dosage. Finally, I think you should have a specific group for the analysis (adult, non-trainined, trained, atlhetes etc) or a region of the body, or even a specific exercise. The work is very wide and without a direction it may be unlikely to be carried out. 

We have responded to the statistical approach below and made changes in the manuscript. We also address the thought about delimiting the search to the specific comment below, but in this instance of attempting to compute the dose-response function, do not believe this to be appropriate. The size of the expected work should not be considered as a reason for it being unlikely to be carried out. As we have and will indicate, part of the purpose for publishing this protocol is to highlight that we are completing this project, provide the clear protocol that we will be following and to maintain accountability for completing the project. Noting that the time that it takes to complete the project and publish the findings is likely to be greater than one would traditionally expect with a tightly controlled and specified systematic review and meta-analysis.

10. Title: The terms dose, dosing, and dosage are confusing, and only explained along the text. I suggest not leaving them in the title. 

As suggested, we have amended the title to “The influence of considering individual resistance training variables as a whole on muscle strength: A systematic review and meta-analysis protocol”. 

11. Ln 38. You wrote sets x repetitions. I think you would like to say: number of sets x number of repetition. Please clarify.

It appears that this question is specifically for Ln 28. We have amended the abstract as suggested. This now reads as:

Page 2, Lines 21-24:

“To derive RT dose, the following calculation will be implemented: number of sets × number of repetitions × number of exercise × intensity, while RT dosing factors in frequency and RT dosage considers program duration.”

12. Ln. 24. I did not find the Propero ID (https://www.crd.york.ac.uk/prospero/) by the number provided. 

When we originally submitted the protocol manuscript, we had submitted the protocol for registration with PROSPERO. Since then, PROSPERO identified that we were not investigating a clinical population and therefore decided to cancel the registration, . To clarify, the methods presented in this manuscript were not identified as an issue, only that we were not investigating a clinical population. Our protocol has since been registered with The Open Science Framework (OSF). The information has been revised in the abstract and methods. 

Page 2, Lines 18-19: 

“(doi.org/10.17605/OSF.IO/EH94V) systematic review with meta-analysis...” 

Page 10, Line 205-206: 

“This systematic review protocol has been prospectively registered with Open Science Framework (OSF),[51]...”

13. Ln 30 The word style seems to be different.

We have checked the manuscript to correct any differences in font size or style and do not believe any differences remain.

14. Ln 33. The pooled standardised mean differences of what?

As suggested we have amended the sentence to the following:

Page 2, Lines 28-29:

“The pooled standardised mean differences for muscle strength will be estimated using DerSimonian and Laird random effects method.”

15. Ln 34-37. It is necessary to describy what are you comparing. 

We have clarified what we are comparing in the proposed systematic review and meta-analysis. This now reads as:

Page 2, Lines 29-33:

“Linear and non-linear dose-response relationships will be estimated by fitting fixed effects and random effects models using the one-stage approach to evaluate if there is a relationship between exercise dose, dosing and dosage and the effect on muscle strength. Maximised log-likelihood and the Akaike Information Criteria will be used to compare alternative best fitting models.”

16. Ln 37-39. Why do you think the proposed variables are enough to compose the best prespriction for strength increase? Previous studies indicated the strength increase is specific to the range of motion training (https://pubmed.ncbi.nlm.nih.gov/23604798/;
https://pubmed.ncbi.nlm.nih.gov/33977835/;
https://pubmed.ncbi.nlm.nih.gov/31230110/). However, nothing is said about the range of motion. If the strength increase is dependent of the range of motion trained, why this variable was not include or even mentioned?

The descision to incorporate the number of sets, the number of repetitions, intensity, frequency, and duration is due to the already established guidelines. We believe that taking into account the above variables will provide an accurate account of the total dose, dosing, and dosage of a resistance training program, as these variables can be quantified in both relative (arbitary units) and absolute (kilograms or pounds) terms. For other variables like range of motion; which indeed has shown to influence muscle strength, further investigation is needed to determine how muscle strength is impacted throughout the various ranges of motion for each exercise and muscle group. Although research suggests that range of motion is important, all possible ranges have not yet been identified and it is not clear if or how this effects the total volume of work. On this basis, we do not believe that including range of motion in the dose calculations for this study will add substantial benefit. Furthmore, the method in which range of motion is assessed is often approximated using a goniometer or visual observation. It is also reliant on the participant being consistent with their range of motion for each repetition throughout the intervention, and that the researcher can assure that the range of motion has been adhered to throughout the intervention. If range of motion was incorporated as one of the variables for dose calculations, it is likely that few studies would provide the specific range of motion for each exercise. Consequently, this would mean that a study would not be quantifiable and therfore excluded from the meta-analysis; severely limiting the available data and potentially calculating a dose-response effect unlikely. Furthermore, it is hoped that the application of these equations can be easily used within practical settings. As such, we believe that including range of motion as one of the variables in the equation would not be practical for most health and fitness professionals.

However, we appreciate that there are variables that we have not considered and have added text to identify that we have selected variables based only on those recommended and outlined in resistance training guidelines. We have also added text that identifies the potential role of other variables (including range of motion). We have amended the text to be more conservative about how the findings might be used.

17. Ln 51.” Resistance training guidelines have been developed to guide”: repetitve terms. 

We have amended this sentence to the following:

Page 3, Lines 45-46: 

“Resistance training guidelines have been developed to direct the attainment of muscle strength, hypertrophy, and power.[6, 7]”

18. Ln 55. why the other variables were not cited?

Thank you for identifying this, this appears to be an issue with EndNote. This has now been corrected, and we have also included repetition cadence in the variables. 

19. Ln 56. The reference cited only informs about volume and intensity. What about the other variables mentioned? They shoud be covered by other references. 

As addressed in response to comment 17, we have added additional references for the other variables, and included repetition cadence. Please see Page 3, Line 50.

Page 3, Lines 46-50. 

“These guidelines have been adapted for individuals across the lifespan, from children to older adults,[8-11] and encompass the manipulation of training variables; exercise selection and order, repetition cadence, range of motion, rest intervals, weight load, frequency, sets and repetitions, to specifically stimulate mechanical and metabolic stress.[12-18]”

20. Ln 56-58. You present a problem that you are not able to solve by the analysis offered, as this prescription requires the analysis of other variables such as execution speed, range of motion and pause. I recommend you be more conservative.

We understand that range of motion and pause, as well as repetition cadence are important variables in exercise prescription. However, at this point do not believe that there are accurate quantification methods being utilised and it is unclear how these would contribute to training volume. As previously addressed, we believe that range of motion should not be included in the dose calculation and that range of motion is often assessed as an approximation and not utilising a machine or exercise that has in place the specific range of motion being investigated. In addition, if range of motion was incorporated as one of the variables for dose calculations, it is likely that few studies would provide the specific range of motion for each exercise, which may severely limit the data necessary to calculate a dose-response effect. With regards to execution speed, or what is better known as repetition cadence, we believe that this is a variable that can be quantified to some extent. We have not included this variable in the proposed dose calculations as studies that specify a particular repetition cadence often don’t specify if this cadence was performed for all exercises, and if this cadence was performed during maximal strength testing, hence questioning the validity of measurement. If the maximal strength test was not performed at the specified repetition cadence, then it is possible that the maximal strength would not accurately reflect the relative intensity of the exercise prescription, and consequently impacting the total dose calculations. In addition, although repetition cadence can be quantified in seconds, the contributions of concentric and eccentric loading to volume are presently unquantifiable.

Again we have identified that we are not considering all possible training variables and have amended the text to be more conservative in our possible explanations and uses. Further we have refined the statement of the problem to.

Page 3, Lines 56-59:

“It is not yet known how these specific variables (sets, repetitions, intensity, frequency, and duration of a program) contribute to the total dose of RT as their interaction has not yet been determined.” 

21. Ln 66 You describe a problem (which variable is most important for maximal strength increase...) that you won't be able to answer. I suggest rephrasing the sentence.

The sentence specifies that based on the literature, there is no clear resistance training variable that is responsible for muscle strength development. We then expand and propose that there may not be a single variable that is primarily responsible, but rather a combination of variables. We have clarified in the manuscript the specific variables that are being referred to as not all training variables can be investigated. This sentence has been rephrased to the following. 

Please see Page 4, Lines 67-70:

“Therefore, it remains unclear whether a specific training variable (e.g. the number of sets, the number of repetitions, intensity, frequency, or duration) is more important than another, or if the interaction of variables results in larger muscle strength adaptations.”

22. Ln 71-73. The analysis of volume x intensity allows to understand the dose of the training section. The analysis of volume x intensity x frequency allows us to understand the weekly dose, but the analysis of these variables with duration would allow which interpretation? nothing was said about it?

As suggested, we have amended this section, and is as follows.

Page 4, Lines 71-77:

“There are three potential ways in which the variables are likely to interact with one another; 1) RT dose considers the interaction of RT volume (number of sets x number of repetitions x number of exercises) and intensity, allowing an individual RT session to be considered; 2) RT dosing considers the interaction between RT volume, intensity, and frequency, allowing a week of RT sessions to be considered; and 3) RT dosage considers the interactions between RT volume, intensity, frequency, and the duration of the program intervention, allowing all RT sessions throughout the intervention period to be considered.[28, 29]”

23. Ln 76. “must consider the duration of a program”. Why? Please, you need to clarify these issue. What about other variable that were left out, such as workload? (https://pubmed.ncbi.nlm.nih.gov/21113614/)

Workload is often calculated as: sets × repetitions × intensity, however different studies have clarified this term differently. Often in literature it is not clarified if the workload calculated is for all exercises prescribed in the intervention or for the exercises where maximal strength is being assessed. In this manuscript, we propose that resistance training dose should be calculated as: the number of sets × the number of repetitions × the number of exercises × intensity. This incorporates the calculation of ‘workload’ while providing clarity on the number of exercises the calculation is referring to. It is important to note that this calculation only considers the potential effect of a single resistance training session, and that in exercise prescription multiple training sessions are performed per week and can continue for many weeks, months, and years. As such, when the frequency (the number of sessions per week) is considered, this calculation considers the potential effect of resistance training over the course of a single week. When the duration (the number of weeks the intervention lasts) is considered, this calculation considers the potential effect of the whole resistance training program. We appreciate the comment and have clarified why resistance training duration should be considered, we hope this is suitable. The amendment is provided below.

Page 4, Lines 77-80:

“Although Price et al.[29] considered the number of sets, repetitions, load, and frequency in their evaluation of RT dosage, this fails to consider the duration of a program which would incorporate all training sessions from the beginning until the end of an intervention. “ 

24. Ln 80 a 137. The concepts of each variable could have been summarized in two paragraphs, and then, the reasoning of the neurophysiological mechanisms that support the importance of analyzing the interaction of the variables presented in the force response could have been presented.

We agree that the concepts of each variable could have been summarised in two paragraphs, however we feel that the expanded overview of each is necessary to define the variables and potential interaction around dose, dosage and dosing. However, we also agree that the neurophysiological mechanisms that might support the interaction of variables was overlooked in this approach. Therefore, we have chosen to integrate the potential mechanisms for each variable rather than presenting a separate section on the mechanisms. Partly this is because the mechanisms of increased muscle strength remain inconclusive and difficult to define. We hope that this approach satisfies the reviewer.

25. Ln 111. “It is also important to consider the influence of RT frequency and duration” . Why it is importante to consider? Please clarify

To be more concise we have removed this sentence as it did not add any additional material to this manuscript. The importance of considering RT frequency and duration is described in the following section of the manuscript, which can be found on Pages 6-8, Lines 128-164.

26. Ln 115-117. No reference was given.

We apologise for missing this. A reference has now been added. Please see Page 6 Line 130.

27. Ln 122. It is important to metion that it is account only for the same muscle group. 

We have corrected this, and specified that the resistance training frequency was recommended for individual muscle groups. This sentence has been amended.

Page 7, Lines 135-138:

“This RT frequency has been reaffirmed by Feigenbaum et al.[7] who recommend a 48 hour rest between concurrent training sessions, which corresponds to three training sessions per week for individual muscle groups.”

28. Ln 135-137. Little information was given to “Duration” compared to the other variables. 

The information around “duration” is limited but reflects the amount of research that has been conducted on this topic. To the best of our knowledge, no original research study or systematic review and meta-analyses has specifically attempted to investigate the influence of resistance training program duration on muscle strength. Within resistance training guidelines, there is also no recommendation for program duration. This is possibly due to training being recommended to be life-long. However, this is a training variable that could contribute to strength development particularly in specific populations (i.e. untrained). While it could be considered that the program duration should be 4-6 weeks to be in line with broad recommendations about when overload should be applied, there is no good quality evidence to support this. 

Inadvertently this variable has been examined through the re-testing of maximal strength throughout an intervention, with strength seemingly plateauing as duration is extended, but limitations in the study design used do not enable conclusive interpretations. As resistance training programs can vary widely from publication to publication either through sample size, population characteristics, program intervention, and study design, etc., it is not yet possible to transfer results from single or small numbers of studies into real-world applications. We have added some text to highlight this in the manuscript (Pages 7-8 Lines 151-164).

29. Ln 149. The question presented has not yet been answered and will not be answered by the study. I see no justification for your permanence. Perhaps present a question closer to the question you want to answer.

Although the question posed is an appropriate question, on reflection we recognise the challenges with being able to conclusively answer that question. We have amended the problem question to the following. 

Page 9, Lines 180-181:

“Does the interaction of specific resistance training variables to quantify the amount of training completed influence the dose-response function for muscle strength.“

We feel that this question better reflects the studies proposed aims, and that this question can be answered with the study design we propose.

30. Ln 154-158. Please clarify how the dose, dosing and dosage could influence the development of muscle strength.

It is plausible that the interaction of different resistance training variables in the form of dose, dosing and dosage will positively combine neurophysiological mechanisms for the development of muscle strength to lead to larger effects. We have added some text to highlight the potential mechanisms that could be induced might be summative and lead to a positive interaction with larger (magnified) dose-responses. 

Page 9, Lines 189-199

“It is important to delineate how the dose (number of sets × number of repetitions × number of exercises × intensity), dosing (sets × repetitions × exercises × intensity × frequency) and dosage (sets × repetitions × number of exercises × intensity × frequency × duration) influences the development of muscle strength, and if a primary driver of this outcome can be identified. These combinations of variables have the potential to induce different neurophysiological mechanisms including motor unit recruitment and firing, motor learning (specific fibre types), agonist-antagonist co-activation, metabolic stress, and increased cross-sectional area. However, it is unknown if these variables and potential mechanisms indeed interact or are summative to induce larger increases in muscle strength. Given that changes in training volume, training intensity and intervention duration have been identified as important factors by Borde et al.[50] and are likely to induce different mechanisms…”

31. Ln 174. “All” articles prior to july 2021? I think you should delimitated a specific group (i.e., https://www.ncbi.nlm.nih.gov/pmc/articles/PMC4656698/), or a body member, or an specific exercise.

We understand and appreciate the reviewers concern about the size of this review, however we believe that it is appropriate not to delimit the search in this instance to ensure that a large enough sample is available to generate an accurate dose response curve. Although we anticipate a high volume of articles, we believe that it is necessary to encapsulate as many manuscripts as possible. We chose not to delimit to a specific population because there is no good evidence to support variation in muscle strength development between populations and we are interested in assessing if the resistance exercise dose, dosing, or dosage required to stimulate muscle strength changes with factors like age. These additional variables can be included in analyses as co-variates and specifically investigated in sub-analyses. Depending on the number of studies, there might be the opportunity to conduct sub-anlyses to see if the dose-response curve is different for specific populations. We chose not to deliminate the whole-body because we believe that this would impact the practical application of our data. For example, the prescription of split upper- and lower-body resistance exercise is common in experienced lifters, however, is not often prescribed for untrained populations. We did not want to deliminate a specific exercise as we believe this would severely limit the data that would be eligible for extraction, this would be particularly evident in studies that include multiple exercises. Further, this would impact the calculation of resistance training dose, as the calculation considers the number of exercises. We understand that the approach that will be undertaken will result in many studies, but we believe that it is necessary to answer questions and we have a specific research team working on this to ensure that it is achievable. In response to the editor also wanting the search to be the most up-to-date as possible, we have made a decision to search up until 1st December 2021. Please see Page 10, Lines 215-216.

32. Ln 178. Which groups?

This sentence was describing the fact that we were considering non-exercise comparators to be able to calculate the size of effect to be considered in the dose-response analysis. We appreciate that this wasn’t expressed as clearly as it could have been. We have made some adjustments to make this clearer.

Page 11 Lines 221-225:

“Muscle strength data from eligible intervention groups will be compared with data from individuals receiving no exercise intervention or a placebo exercise intervention (i.e. stretching and/or mobility exercise) not designed to improve strength. Aerobic exercise only interventions have not been considered appropriate as a comparator to exclude any potential strength benefits from this mode of training, particularly in exercise-naïve populations.”

33. Ln 235. Please, give an example for better understanding. Why the median and not the mean?

For the purpose of consistency between extracting data, we believe that extracting the median is more appropriate given that intensity can widely range in exercise prescription (e.g. 20%1RM to 100%1RM). From a provided range, it is not appropriate to calculate the mean since the mean is determined from a series of exact values. Hence the decision to use the median (central value). In most instances, we think the mean and median would provide essentially the same value. We also acknowledge that the progression of intensity over long durations will vary and may not be normally distributed. Moreover, the duration of intensity prescription may not be normally distributed over the course of an intervention. For instance, participants are prescribed 40%1RM for 2-weeks, followed by 4-weeks at 60%1RM, 8-weeks at 80%1RM, and 3-weeks at 90%1RM. Therefore, if the mean was used instead of the median, it is possible that intensity could be under- or over-estimated. The median on the other hand provides the central tendency for skewed number distribution, which given the previous example, would provide a more accurate account of intensity. While we acknowledge that either approach is likely to contain some degree of error, we feel that the most appropriate approach is to determine the median.

We have provided an explanation why we have chosen the median, as the mean is often reported in literature. The followed was added to the manuscript.

Page 14, Lines 273-277:

“The median was chosen as wide ranges in resistance training intensity (i.e. 20% 1RM to 100%1RM) can be applied, along with variation in how it is progressed throughout an intervention. If the mean was to be used instead of the median, it is possible that intensity could be under- or over-estimated. Utilising the median will provide the central tendency for skewed number distribution.”

34. Ln 245-248. Why 10%? What is the rationale for this procedure?

We are anticipating that this study will identify hundreds of eligible articles for data extraction. Although it is recommended by Cochrane that data are extracted independently by more than one person, it is also rare for systematic reviews to include the number of articles that we are expecting. Therefore, we have made a pragmatic decision to check accuracy of a single data extractor initially in 10% of included articles. As this is a process to ensure integrity of data, if there is any variation in data extraction outcomes, data from all articles will be completed independently by two investigators. 

35. Ln 268. No answer is absolute. Each study adds knowledge on the subject and its contribution cannot be disregarded. Please re-do the sentence.

Our intention certainly was not to disregard any study or findings, but simply to identify that there are limitations in the evidence and that we are proposing a way to potentially address some of these limitations. We have amended this sentence and it now reads as:

Page 16, Lines 331-336:

“Although previous attempts to evaluate the dose-response of individual RT variables using standardised mean effects have identified key variables of interest, there has not been a consistent approach to determining the overall dose, dosing and dosage obtained through the interaction of variables.[50] The approach proposed will utilise dose as a continuous variable instead of categorising variables/outcomes to identify if one category is different to another.”

36. Ln 269. “Instead, previous analyses have simply categorised variables/outcomes to 270 identify if one category is different to another “ . As it is, it gives the impression that the previous work was reduced and contributed little. I believe that the phrase could be redone by valuing the previous work and at the same time adding actions that would complement the previous initiative.

It was not our intention to devalue another authors work. We have amended this section and further expanded from the above amendment. This now reads as.

Page 16, Lines 331-336:

“ Although previous attempts to evaluate the dose-response of individual RT variables using standardised mean effects have identified key variables of interest, there has not been a consistent approach to determining the overall dose, dosing and dosage obtained through the interaction of variables.[50] The approach proposed will utilise dose as a continuous variable instead of categorising variables/outcomes to identify if one category is different to another.”

37. Ln 271-272. You are disregarding the other variables. I think the phrase would be better if you commented only for the effect of the interaction between the variables that are being investigated.

We acknowledge that we are not considering all potential variables and this is partly because it is currently not clear how to best quantify some of these. To address this, we have amended the sentence to specify the training variables that are being analysed in the proposed study. This now reads as:

Page 16-17, Lines 336-340:

“By gathering, analysing, and synthesising the information about the number of sets, number of repetitions, number of exercises, and RT intensity, frequency, and duration of the RT program on a continuous scale and evaluating if the interaction of these variables influences muscle strength differently, this study might offer new directions for practice and future research.”

---

## [Decision Letter · Decision Letter 1]

3 Jan 2022

The influence of considering individual resistance training variables as a whole on muscle strength: A systematic review and meta-analysis protocol.

PONE-D-21-23722R1

Dear Dr. Philip Lyristakis,

We’re pleased to inform you that your manuscript has been judged scientifically suitable for publication and will be formally accepted for publication once it meets all outstanding technical requirements.

Kind regards,

Samuel Penna Wanner, Ph.D.

Academic Editor

PLOS ONE

Additional Editor Comments (optional):

After reading the revised study protocol, the response to the reviewers’ concerns, and considering the first reviewer’s report, I believe that the study protocol has merit and represents an initial step for an exciting and relevant systematic review and meta-analysis. The authors were highly responsive and adequately addressed every comment made by the two reviewers. Congratulation on this! Of note, the second reviewer declined to evaluate the revised study protocol, possibly because he initially recommended its rejection. However, I judge that the authors have succeeded in addressing all issues raised by this reviewer. It should also be highlighted that the authors have answered criticisms (some were strong ones) in a very polite and constructive way. Therefore, taking the above information into account, I recommend accepting the study protocol.

I am looking forward to reading the systematic review and meta-analysis when available.

Finally, If the authors get a chance to make final amendments to the manuscript, please consider:

a- To split the first paragraph of the introduction into two paragraphs. Please also consider flipping the order of the last two sentences of this paragraph.

b- To insert a comma before the word “including” (line 194).

c- To insert the word “also” between the word “are” and “likely” (line 346).

Reviewers' comments:

Reviewer's Responses to Questions

**Comments to the Author**

1. Does the manuscript provide a valid rationale for the proposed study, with clearly identified and justified research questions?

Reviewer #1: Yes

2. Is the protocol technically sound and planned in a manner that will lead to a meaningful outcome and allow testing the stated hypotheses?

Reviewer #1: Yes

3. Is the methodology feasible and described in sufficient detail to allow the work to be replicable?

Reviewer #1: Yes

4. Have the authors described where all data underlying the findings will be made available when the study is complete?

Reviewer #1: Yes

5. Is the manuscript presented in an intelligible fashion and written in standard English?

Reviewer #1: Yes

6. Review Comments to the Author

You may also provide optional suggestions and comments to authors that they might find helpful in planning their study.

Reviewer #1: I am satisfied with the answers provided. In my opinion the quality of the work increased after the review. No additional comments

7. PLOS authors have the option to publish the peer review history of their article (what does this mean?). If published, this will include your full peer review and any attached files.

Reviewer #1: **Yes: **Lucas Rios Drummond

---

## [Editor Report · Acceptance letter]

7 Jan 2022

PONE-D-21-23722R1 

The influence of considering individual resistance training variables as a whole on muscle strength: A systematic review and meta-analysis protocol. 

Dear Dr. Lyristakis:

I'm pleased to inform you that your manuscript has been deemed suitable for publication in PLOS ONE. Congratulations! Your manuscript is now with our production department. 

Kind regards, 

on behalf of

Dr. Samuel Penna Wanner 

Academic Editor

PLOS ONE